# Evaluating spoken language as a biomarker for automated screening of cognitive impairment

Maria R. Lima [1,2] ✉, Alexander Capstick [2,3], Fatemeh Geranmayeh [3,4], Ramin Nilforooshan[2,5], Maja Matarić[6], Ravi Vaidyanathan [1,2,7] & Payam Barnaghi [2,3,7]

## Abstract

**Background** Timely and accurate assessment of cognitive impairment remains a major unmet need. Speech biomarkers offer a scalable, non-invasive, cost-effective solution for automated screening. However, the clinical utility of machine learning (ML) remains limited by interpretability and generalisability to real-world speech datasets. **Methods** We evaluate explainable ML for screening of Alzheimer's disease and related dementias (ADRD) and severity prediction using benchmark DementiaBank speech ($N = 291$, 64% female, 69.8 ± 8.6 years). We validate generalisability on pilot data collected in-residence ($N = 22$, 59% female, 76.2 ± 8.0 years). To enhance clinical utility, we stratify risk for actionable triage and assess linguistic feature importance. **Results** We show that a Random Forest trained on linguistic features for ADRD detection achieves a mean sensitivity of 69.4% (95% confidence interval (CI) = 66.4–72.5) and specificity of 83.3% (78.0–88.7). On pilot data, this model yields a mean sensitivity of 70.0% (58.0–82.0) and specificity of 52.5% (39.3–65.7). For prediction of Mini-Mental State Examination (MMSE) scores, a Random Forest Regressor achieves a mean absolute MMSE error of 3.7 (3.7–3.8), with comparable performance of 3.3 (3.1–3.5) on pilot data. Risk stratification improves specificity by 13% on the test set, offering a pathway for clinical triage. Linguistic features associated with ADRD include increased use of pronouns and adverbs, greater disfluency, reduced analytical thinking, lower lexical diversity, and fewer words that reflect a psychological state of completion. **Conclusions** Our predictive modelling shows promise for integration with conversational technology at home to monitor cognitive health and triage higher-risk individuals, enabling early screening and intervention.

## Plain language summary

Signs of dementia can manifest in speech years before other clinical symptoms appear. Machine learning methods can detect subtle speech changes to support early screening without increasing clinicians' workload. This study assesses spoken language biomarkers for automated screening of cognitive impairment. We analyse verbal picture descriptions from the benchmark DementiaBank and validate the method on an independent cohort. Our findings demonstrate robust detection of cognitive impairment and prediction of disease severity. We introduce a risk stratification approach to identify individuals at higher risk who require further assessment. Linguistic markers of cognitive decline include frequent pronouns and adverb use, greater disfluency, and less diverse vocabulary. This approach could be integrated with conversational technologies to provide accessible home-based monitoring of cognitive health and support timely interventions.

There is a pressing need for accurate, accessible, and cost-effective risk assessment methods for the early identification of cognitive decline in at-risk groups. Dementia diagnoses are typically made years after symptom onset, missing a crucial therapeutic window that is becoming increasingly important with the recent emergence of anti-amyloid drugs[1]. Traditional Alzheimer's disease and related dementias (ADRD) diagnostic methods rely on identifying fluid biomarkers such as *Tau* and *β*-amyloid related proteins, or neuroimaging techniques such as positron emission tomography and

magnetic resonance imaging[2]. While informative, these techniques are invasive, expensive, and inaccessible for scalable population screening[3]. Furthermore, brain imaging is only useful when signs of neurodegeneration manifest, missing a therapeutic window of opportunity.

Administering neuropsychological tests (NPT) through an in-person interview remains the primary method to evaluate cognitive functions, including attention, memory, language, and visuospatial abilities. However, NPT are limited by clinician availability, are often qualitative in nature, and

[1]Department of Mechanical Engineering, Imperial College London, London, UK. [2]UK Dementia Research Institute, Care Research and Technology Centre, London, UK. [3]Department of Brain Sciences, Imperial College London, London, UK. [4]Imperial College Healthcare NHS Trust, London, UK. [5]School of Psychology, University of Surrey, Guildford, UK. [6]Department of Computer Science, University of Southern California, Los Angeles, CA, USA. [7]These authors contributed equally: Ravi Vaidyanathan, Payam Barnaghi. ✉e-mail: maria.lima@imperial.ac.uk

are susceptible to errors and high inter-rater variability. Furthermore, results can be affected by non-cognitive factors, such as mood disorders and fatigue, and expertise is required when interpreting the results to avoid false positive diagnoses [4].

There has been recent interest in deriving early spoken language features of ADRD as biomarkers, which can be collected in an ecologically valid manner. Increasing evidence suggests that speech and language can be strong predictors of cognitive decline in the early pre-clinical stages of ADRD [5–8]. Neuroimaging studies also indicate that semantic fluency and naming performance are highly correlated with neurodegeneration in the temporal and parietal lobes [9], areas commonly affected in Alzheimer's disease (AD). Changes in acoustic and linguistic characteristics have been linked to cognitive decline, including slower speech rate, more disfluencies (e.g., frequent pauses, hesitations, repetitions), reduced noun use, and increased use of pronouns, verbs, and adjectives [10–13]. Whilst previous studies have primarily focused on analysing speech and language from voice recordings of NPT [14], their use in real-world settings outside clinics in pre-clinical populations remains underexplored. Recent studies have suggested the feasibility of collecting speech via mobile applications and voice assistants to detect mild cognitive impairment (MCI) and ADRD [15–17]. This opens new opportunities for the use of in-home conversational technologies to monitor cognitive health.

Deep learning has been used for automatic feature extraction with pre-trained models for audio and text representation [18]. Recent attempts have explored the potential of large language models, such as BERT and GPT, for assessing cognitive decline [19,20]. These transformer-based models can automatically capture subtle language patterns potentially missed by conventional methods. However, their lack of explainability hinders clinical applicability. An additional advantage of deep learning approaches is the ability to extract multilingual embeddings. This is an area of active research [21] and could help address the limited sample sizes in existing speech datasets for ADRD research.

Following feature extraction methods, emerging evidence supports the feasibility and reliability of machine learning (ML) in detecting ADRD and modelling disease progression. For instance, a logistic regression model trained on embedding vectors from NPT transcripts and demographic data achieved an accuracy of 78.5% and a sensitivity of 81.1% in predicting AD progression within six years [19]. Classifiers trained on conventional acoustic features automatically extracted from speech have shown an accuracy of 71.3% [22]. Similarly, a logistic regression model using acoustic and linguistic features extracted from picture description tasks during NPT has achieved an accuracy of 81.9% in binary AD classification [11]. Beyond ADRD detection, previous literature has also applied explainable ML in three-way classification tasks to differentiate MCI from cognitively normal individuals [23,24].

This study explores predictive models for automated screening of cognitive impairment from speech and language with a focus on clinical utility. Our predictive modelling includes ADRD detection and Mini-Mental State Examination (MMSE) prediction to assess state of cognitive decline. We use speech recordings from two DementiaBank datasets [25] ($N = 291$) and validate the best-performing model on an independent dataset of a different cohort, language, and setting ($N = 22$). We extract acoustic and linguistic features using both conventional and deep learning approaches, prioritising feature interpretability to inform clinicians of linguistic changes indicative of cognitive decline.

With an emphasis on clinical utility, this study introduces the following contributions: 1) *Independent validation using our parameterised model without re-training:* we evaluate the best-performing model unchanged on an independent pilot dataset to assess its generalisability across a different cohort, language, and setting. Results suggest the model captures transferable knowledge and could operate as an out-of-the-box solution. 2) *Actionable triage framework:* we introduce a traffic-light stratification approach that categorises model predictions into clinically actionable low, medium, and high risk of cognitive impairment. This approach aims to enhance clinical utility by reducing false positives and streamlining the triage process. 3) *Fairness and performance across different demographic and cognitive groups:* we report subgroup performance by demographic and MMSE splits, demonstrating parity and highlighting expected uncertainty in MCI cases. Particularly, we consider the fact that women are twice as likely to develop Alzheimer's disease as men [26] and report that our chosen model maintains robust predictions across male/female splits. 4) *Predicted state of cognitive decline for prognostic assessment:* beyond detection, we model the state of cognitive decline by predicting MMSE as a continuous outcome and further validate generalisability of the chosen regressor on the pilot dataset. Given the progressive nature of ADRD, we see particular clinical utility in using speech to analyse trajectories of decline. This approach provides a foundation for monitoring and prognostic assessment from longitudinal speech.

## Methods

In this section, we describe the speech datasets used to train and evaluate our models, the ML pipeline, methods for linguistic feature extraction, and the risk stratification approach to enhance clinical utility. We consider the following ADRD severity groups based on MMSE scores according to the UK National Institute for Health and Care Excellence dementia guidelines [27]: cognitively normal (CN) (26, 30], MCI (20, 26], moderate dementia [10, 20], severe dementia [0, 10) (following interval notation).

### Speech datasets

We used the ADReSSo dataset from DementiaBank [25] to train and evaluate our models. This dataset included spontaneous speech recordings produced by CN participants and people with an ADRD diagnosis, who were asked to describe the Cookie Theft picture during NPT [28] (see Supplementary Table 1). ADRD diagnosis here describes a clinical diagnosis of Alzheimer's dementia in the absence of fluid and imaging biomarker confirmation. Therefore, the label is likely to include all-cause dementias. The recordings were acoustically pre-processed with noise reduction and volume normalisation. The dataset contained 237 speech samples (5 h) with a 70:30 train-test ratio balanced for demographics.

To verify the generalisability of our best-performing models for ADRD detection and MMSE prediction, we externally validated them with two datasets beyond the ADReSSo held-out set. We used the Lu corpus from DementiaBank as an external test set [29]. This dataset comprised 54 speech samples (1 h) from the same picture description task with binary labels for CN participants and those with an ADRD diagnosis.

Separately, we collected an additional speech dataset from 22 older adults (46 min) living in retirement homes, who completed the same verbal picture description task. Our dataset includes both English and Spanish speakers. Although participants did not have a dementia diagnosis, we considered two cognitive groups: CN and those with mild to moderate cognitive impairment, using standard MMSE cutoff of 26 as suggested in previous work [30]. We refer to this new dataset collected in-residence from older adults as the pilot study. Participants in the pilot study self-reported their races and ethnicities: 14 African-American, 7 white, 8 Hispanic, and 1 participant did not disclose. Table 1 summarises the demographic characteristics of each study cohort used for training, testing, and the pilot data used for independent testing across a different cohort, language, and setting. Note that the additional DementiaBank test set does not provide MMSE scores, so this dataset was not used for the severity prediction modelling.

### Acoustic and linguistic features

We extracted acoustic and linguistic features using both natural language processing (NLP)-based methods for interpretable features and pre-trained deep learning models. We also analysed three multimodal feature combinations using early fusion methods. The ML pipeline for feature extraction and predictions is illustrated in Supplementary Fig. 1. The acoustic and linguistic features extracted, which were used as input to various ML models, are detailed in Supplementary Fig. 2.

Transcripts were generated from each audio file (i.e., one per participant) using OpenAI's Whisper [31] for automatic speech recognition (ASR) (model *medium.en*, version 20240930). Given the high Spearman's rank

**Table 1 | Demographic characteristics of the study cohorts**

|  | Training | Test | Additional Test | Pilot Study |
|---|---|---|---|---|
| Total | 166 | 71 | 54 | 22 |
| Age (years) | 68 (6.8) | 67.3 (6.9) | 79.3 (9.7) | 76.2 (8) |
| Sex (% male) | 34% | 38% | 41% | 41% |
| MMSE | 22.9 (7) | 23.9 (6.6) | – | 24.9 (3.9) |
| Cognitive group | 79 CN, 87 AD | 35 CN, 36 AD | 27 CN, 27 ADRD | 8 CN, 14 MCI |
| Language | All English | All English | All English | 14 English, 8 Spanish |

Mean (standard deviation) is reported for age and MMSE.

correlation (mean $r = 0.98$, SD = 0.03, $p < 0.05$) between linguistic features manually transcribed from participant-only and combined speaker data across the 237 DementiaBank audio files, we decided to proceed with the remainder of the analysis without automatic speaker diarisation, which proved unreliable in accurately separating participant and administrator speech. From the transcripts obtained with ASR, we extracted linguistic features using two methods: 1) token embeddings, using transformer-based pre-trained language models to create a 1536-dimensional vector representation for each participant's transcript (OpenAI's GPT embeddings); 2) NLP to extract linguistic features, mostly lexical-semantic. The latter allows us to train ML models with clinical applicability by providing interpretable insights into the linguistic patterns that contribute to model predictions. This transparency facilitates more informed clinical decision-making in analysing what attributes of speech and language are indicative of cognitive decline.

We computed five lexical-semantic features, including type-token ratio corrected for text length, Brunet Index, Honore Index, propositional idea density, and consecutive duplicate words. These were combined with lexical and semantic psycholinguistic features extracted using Linguistic Inquiry and Word Count (LIWC), a method that counts words in psychologically meaningful categories[32] (see Supplementary Table 2). We used the LIWC-22 English-only dictionary as it provides a more comprehensive and diverse vocabulary compared to older multilingual versions. Previous studies using LIWC have shown its ability to characterise language in patients with mental and neurological disorders[33,34] and loneliness among older adults[35]. After pre-processing and selection of LIWC subcategories, a 100-dimensional vector was extracted from each participant's transcript. To maintain consistency in linguistic feature extraction, we applied GPT-4o translation to Spanish transcripts before extracting English-based LIWC features, ensuring an end-to-end pipeline from data collection through pre-processing to analysis. We evaluated the most important features influencing predictions using SHapley Additive exPlanations (SHAP)[36]. This method calculates the contributions of individual features to risk scores, providing explainable predictions.

**Predictive modelling**

We performed an ADRD detection task and an MMSE severity score prediction task. For the first, we tested Logistic Regression (LR), Support Vector Machine (SVM), Random Forest (RF), Multilayer Perceptron (MLP), and Extreme Gradient Boosting Decision Tree (XGBoost) models at detecting ADRD from spontaneous speech. For the second task, we tested Ridge Regression (RR), Support Vector Regression (SVR), Random Forest Regressor (RFR), MLP Regressor, and XGBoost Regressor models in predicting MMSE scores. Hyperparameters were tuned using 10-fold cross-validation (CV). We verified the chosen model was well-calibrated before testing (see calibration curves in Supplementary Fig. 3), and performance metrics were averaged across all validation folds. We evaluated the best model on the ADReSSo held-out test set (comprising 30% of the data) that

was not used during model development, as well as on an independent pilot dataset.

**Statistics and reproducibility**

Bootstrapping was used to estimate performance variability on the test set with 10 bootstrap repeats, with each run using a bootstrap sample from the training set to ensure reproducibility. Classification performance was evaluated using sensitivity, specificity, receiver operating characteristic area under the curve (ROC-AUC), and accuracy. We selected ROC-AUC as the primary evaluation metric as it is based on the predicted probability scores, providing a comprehensive assessment of the model's ability to distinguish true ADRD cases while minimising false positives across all classification thresholds. Regression performance was measured with mean absolute error (MAE) and root mean square error (RMSE). Definitions of evaluation metrics are presented in Supplementary Materials. Non-parametric Spearman's rank correlation tests were used to evaluate associations between word transcription type, linguistic features, and cognitive scores, with $p < .05$ considered statistically significant. All analyses were performed in Python v3.12. The scikit-learn, NumPy, PyTorch, and SciPy, and spaCy libraries formed the core of the ML pipeline.

**Model and feature selection**

We evaluated each classifier and regressor on the selected acoustic and linguistic feature sets, as described in Section Acoustic and Linguistic Features. We selected the model achieving the highest ROC-AUC on the validation set for further comparison. The best-performing model using linguistic features was selected for final analysis based on both its predictive performance and the interpretability of features for clinical utility.

**Stratification of risk scores for clinical utility**

To enhance clinical utility, we stratified prediction scores from the best-performing model into three risk groups: Green (low risk), Amber (medium risk), and Red (high risk). The thresholds were determined via 10-fold stratified cross-validation on the validation set. By varying the thresholds for the Amber, Green and Red groups, we could adjust sensitivity and specificity for the different risk groups. We varied the thresholds with a resolution of 10%, and evaluated performance metrics for the Green (positive prediction) and Red (negative prediction) groups on the validation set. Following a selective classification approach[37], we excluded the Amber group, which represents increased model uncertainty. We optimised the coverage of Green and Red zones for higher ROC-AUC, as well as jointly increasing sensitivity and specificity using Yoden's J index[38]. This approach aims to enhance clinical utility by prioritising more confident predictions in the Green and Red groups, which can streamline triage processes and better inform clinical decisions by identifying individuals at higher risk of ADRD. Given the small size of our dataset, when similar results were obtained for different thresholds on the validation set, we selected smaller thresholds to prevent overfitting.

**Ethics statement**

The Alzheimer's Dementia Recognition through Spontaneous Speech only (ADReSSo) data and Lu corpus are available via DementiaBank[29], supported by NIH-NIDCD grant R01-DC008524. Ethical approval for the pilot study was provided by the University of Southern California Review Board UP-24-00154. All participants signed informed consent prior to the study.

**Results**

**Model performance in ADRD detection**

We analysed the effectiveness of LR, SVM, RF, MLP, and XGBoost in detecting ADRD (see details in Supplementary Materials) from the extracted acoustic and linguistic features, using both NLP-based methods to obtain interpretable features and pre-trained deep learning models. We found that models trained with linguistic features achieve higher performance than those using acoustic features for ADRD classification

**Table 2 | Best-performing models using linguistic features for ADRD detection**

| Model-Features | | Sensitivity | Specificity | ROC-AUC | Accuracy |
|---|---|---|---|---|---|
| MLP-GPT | Validation | 79.3 (13.5) | 76.2 (18.1) | 87.5 (7.7) | 77.7 (12.3) |
| RF-NLP | Validation | 78.8 (16.7) | 72.1 (13.4) | 83.5 (8.9) | 75.3 (9.4) |
| | Test | 69.4 (66.4–72.5) | 83.3 (78.0–88.7) | 85.7 (83.8–87.6) | 76.5 (74.4–78.6) |
| | External test | 80.0 (77.2–82.8) | 74.1 (69.6–78.6) | 84.6 (82.8–86.4) | 77.0 (74.6–79.5) |
| | Pilot study | 70.0 (58.0–82.0) | 52.5 (39.3–65.7) | 65.4 (54.9–70.1) | 63.6 (54.7–72.6) |

Evaluation metrics include sensitivity, specificity, ROC-AUC and accuracy, reported as mean (standard deviation)% for the 10-fold CV, and as mean (95% confidence interval (CI))% for the test set with 10 bootstrap repeats.

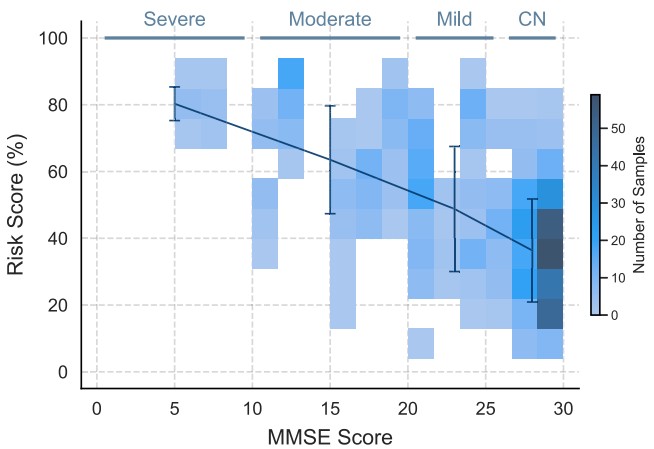

**Fig. 1 | Predicted positive cases per cognitive group.** Distribution of predicted probabilities for ADRD detection on the test set across MMSE scores, considering 10 bootstrap repeats. Central points represent the mean predicted probability per cognitive group (CN, mild, moderate, severe) and error bars show the standard deviation across all observations within each cognitive group. Lower MMSE values indicate worse cognition.

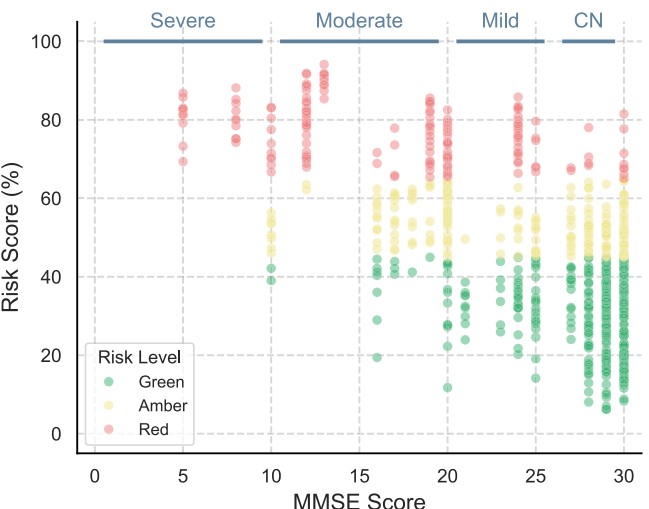

**Fig. 2 | Risk level distribution by MMSE scores.** Distribution of the Green, Amber and Red risk groups across each MMSE score on the test set for the RF model using interpretable linguistic features. The prediction results are reported considering 10 bootstrap repeats.

(see Supplementary Table 3). The best-performing model was an MLP using Generative Pre-trained Transformers (GPT) embeddings (referred to as MLP-GPT), as described in Table 2. However, the RF model of 50 decision trees with depths of 16 (chosen through hyperparameter optimisation) using linguistic features from NLP (referred to as RF-NLP) achieve a comparable mean ROC-AUC on the validation set with only 4% lower performance than MLP-GPT, while offering more interpretability and efficiency. This is partly explained by the RF-NLP model using a more compact (size 100) and interpretable linguistic feature set compared to the GPT embeddings (see Section Acoustic and Linguistic Features). Additionally, this model does not require the use of pre-trained transformer-based models, which lack explainability. These more explainable features can better inform clinicians of linguistic changes indicative of cognitive deterioration. Therefore, considering our study's focus on the clinical utility of automated screening of cognitive impairment, the subsequent results are presented for the RF-NLP model.

On the test set, the model achieves a ROC-AUC, our primary evaluation metric, of 85.7% (95% CI = 83.8–87.6). To further assess model generalisability, we evaluated the RF-NLP model on an additional DementiaBank dataset[29] never seen during model training and used only for a final test. Results suggest good model generalisability on unseen data using the NLP features, with a comparably high ROC-AUC of 84.6% (95% CI = 82.8–86.4). Table 2 presents the performance results on the validation and test sets using linguistic features. In Supplementary Table 4, we report the performance across different demographic groups (sex and age) and verify high demographic parity[39], suggesting our model is a fair classifier.

## Correlation with cognitive scores

Figure 1 shows that the worse the cognitive impairment the higher the predicted positive probability of the RF-NLP model. Although the classifier was trained with binary labels (i.e., not based on severity), the predicted probabilities obtained are correlated with cognitive impairment as measured by MMSE. This property of our modelling is particularly noteworthy as it demonstrates that our model is well-calibrated with both the predicted probability of dementia and its severity, without it being explicitly trained on the latter. Furthermore, the model shows lower confidence and higher variability in its predictions for the mild cognitive impairment group. This is anticipated because individuals with MCI exhibit more subtle changes in language[40] compared to those with more advanced cognitive impairment, increasing model uncertainty in distinguishing between MCI and other cognitive groups. In Supplementary Table 5 and Supplementary Fig. 4 we report model performance results per MMSE group.

## Risk analysis

To enhance clinical utility, we calculated risk thresholds on the validation set that represent minimal (Green), medium (Amber), and high (Red) risk of ADRD (see details in Section Stratification of Risk Scores for Clinical Utility). We selected thresholds [0%, 45%], (45%, 65%], and (65%, 100%] (following interval notation) for Green, Amber, and Red risk groups, respectively. Figure 2 shows the distribution of ADRD risk levels for each MMSE score on the test set. Furthermore, grouping the predictions of Red and Green risk levels following a selective classification approach[37] improves model performance to a mean ROC-AUC of 88.7 (95% CI = 86.2–91.2), sensitivity of 67.6 (95% CI = 62.1–73.2), specificity of 96.7

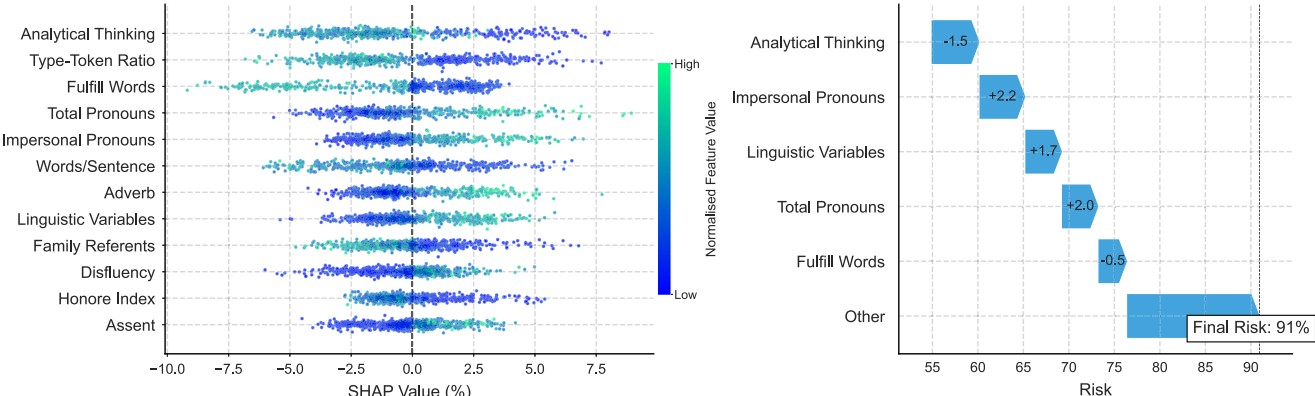

**Fig. 3 | SHAP results.** The left panel shows the feature importance for the top 12 features in the test set and their corresponding feature values from the RF-NLP model. Lower SHAP values suggest reduced predicted probability of ADRD. The colour scale represents the normalised feature value, and the x-axis position indicates the contribution of that value to the prediction. The right panel shows SHAP values for a single positive prediction, illustrating how each feature contributed to model output. Numbers on the arrows correspond to normalised feature values expressed in standard deviations from the mean.

(95% CI = 93.3–100), and accuracy of 83.6 (95% CI = 80.6–86.5) on the test set. Of particular note, when creating the Amber risk group, the model better captures true negative cases (i.e., CN). Such risk stratification approach aims to reduce false positives, streamline triaging, and inform translational pathways for clinical adoption. For instance, high-risk cases can be acted on immediately with efficient allocation of medical resources, while uncertain cases can be flagged for further diagnostic workup.

### Linguistic feature importance

The results obtained with SHAP analysis are shown in Fig. 3. This analysis indicates that lower predicted probability of ADRD (i.e., true negative cases) is associated with higher levels of analytical thinking, higher lexical diversity, more frequent use of fulfil words, i.e., words expressing satisfaction or completion (e.g., 'enough', 'complete', 'full'), greater average words per sentence and more frequent references to family-related words. Conversely, SHAP analysis indicates that more frequent use of pronouns, particularly impersonal pronouns (e.g., 'that', 'it', 'this'), adverbs (e.g., 'there', 'so', 'just'), and lower Honore Index values – indicative of reduced vocabulary richness and increased repetitiveness – contribute to higher ADRD predicted probability (i.e., true positive cases). Additionally, higher linguistic disfluency – a normalised feature that aggregates the frequency of LIWC-defined categories nonfluencies (e.g., 'uh', 'um', 'oh'), fillers (e.g., 'you know', 'so'), and assent words (e.g., 'yeah', 'ok') – is linked with higher predicted probability of ADRD.

The increased reference to family-related words by CN participants suggests greater cognitive inference ability and more detail provided in picture descriptions. Note that the Cookie Theft picture illustrates family activities and actions in a kitchen setting. Furthermore, the increased frequency of pronoun and adverb usage among participants with greater cognitive impairment may suggest difficulty in retrieving specific terms, relying on a more restricted and less diverse vocabulary to describe the scene. This use of language may also indicate prolonged cognitive processing times, increased hesitations, word-finding difficulty and reduced linguistic complexity.

We also examined single predictions to understand contributions to a specific risk score. Figure 3 shows an example of a correct positive prediction with 91% probability driven by lower analytical thinking, more frequent impersonal pronouns and overall linguistic variables, and decreased use of fulfill words. Descriptions of relevant linguistic features are included in Supplementary Table 2. Further examples of individual predictions are shown in Supplementary Fig. 5.

### Model performance in MMSE prediction

We examined RR, SVR, RFR, MLP Regressor, and XGBoost Regressor in predicting MMSE scores (see details in Supplementary Materials). The RFR

using NLP features (RFR-NLP) emerges as the best-performing model, with a minimum of two samples per leaf selected through hyperparameter optimisation. This model achieves a mean absolute MMSE error of 3.7 (95% CI = 3.7–3.8) on the test set. Table 3 presents model performance results and Fig. 4 shows the average MAE per participant in the different cognitive groups for the test set. The model performs better for higher MMSE scores, which could be due to the uneven distribution of the available data, with severe cognitive impairment representing only 5.7% of the test set. The higher MAE observed for the severe cases may also reflect noise in the linguistic features used by the ML model for predictions. We found a moderate positive correlation (Spearman's rank $r = 0.53$, $p < 0.05$) between the proportion of participant-only transcribed speech and MMSE scores, indicating that those with worse cognition require more intervention from the administrator during the task (see Supplementary Table 6). Results of the other evaluated models are reported in Supplementary Table 7.

### Pilot study

We extended our analysis to an independently collected speech dataset, applying the model without re-training or fine-tuning, to assess the generalisability of our ML approach across a different cohort, language, and setting. We collected multilingual speech samples from 22 older adults living in retirement homes who completed the same picture description task in English or Spanish. The RF-NLP model achieves a ROC-AUC of 65.4 (95% CI = 54.7–72.6) on this new dataset, as reported in Table 2. The lower model performance observed on this pilot dataset, particularly on the ability to detect true negatives (i.e., Specificity), may reflect the predominance of participants in the mild and CN cognitive groups (see details in Table 1). These groups generally exhibit lower predicted probabilities for distinguishing cognitive impairment (see Fig. 1), so their higher representation in the pilot dataset likely contributes to this effect.

Using the same risk thresholds (see Section Risk Analysis) and grouping predictions of Red and Green levels enhances model performance to ROC-AUC of 67.3 (95% CI = 61.4–73.1), with higher Specificity of 73.5 (95% CI = 55.4–91.6) at the expense of lower Sensitivity of 53.0 (95% CI = 37.4–68.7). Additionally, when predicting MMSE scores (severity task), the RFR-NLP model achieves a MAE of 3.3 (95% CI = 3.1–3.5), improving on the results obtained in the DementiaBank test set. Despite the small sample size, this pilot study underscores the potential of using linguistic features elicited during picture descriptions to model the state of cognitive decline, even when collected in real-world settings outside clinics.

### Discussion

Monitoring cognitive health is essential for early screening and timely intervention in both clinical practice and daily care. The 2024 report of the Lancet Commission on dementia prevention, treatment, and care[41]

**Table 3 | Severity prediction results using the best-performing RFR-NLP model**

|  | MAE | RMSE |
|---|---|---|
| Validation | 4.8 (0.5) | 5.9 (0.7) |
| Test | 3.7 (3.7–3.8) | 4.7 (4.6–4.8) |
| Pilot study | 3.3 (3.1–3.5) | 4.2 (3.9–4.4) |

Evaluation metrics include MAE and RMSE, reported as mean (standard deviation) for the 10-fold CV, and as mean (95% CI) for the test set with 10 bootstrap repeats.

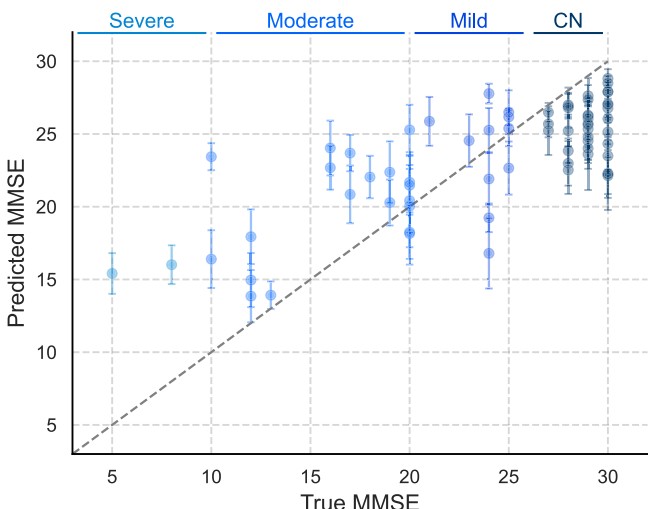

**Fig. 4 | Model performance in severity prediction across cognitive groups.** MAE for predictions on the DementiaBank test set ($N = 71$ participants, each with one MMSE score). Central points represent the mean predicted MMSE across 10 bootstrap repeats for each participant. Error bars show the standard deviation across repeats as an estimate of model uncertainty.

emphasises that timely diagnosis supports the well-being of people living with dementia and their families, facilitates access to services, and ensures that individuals benefit from treatments when they are most likely to be effective. Recent advances in disease-modifying treatments for early-stage AD, including trials of amyloid-$\beta$-targeting antibodies, have shown modest efficacy, creating a therapeutic window of opportunity for intervention, which should follow an adequate diagnosis[42,43]. Early screening can also help to reduce unnecessary hospitalisations and improve overall dementia care. The report further highlights the potential of mobile and wearable devices to detect neurodegeneration through continuous monitoring of physical and cognitive changes. Within this context, speech biomarkers have emerged as a promising approach for scalable, non-invasive, and cost-effective screening of cognitive impairment.

We present an explainable ML pipeline for automated screening of cognitive impairment and ADRD severity prediction from spoken language, placing emphasis on a path to clinical utility. We used DementiaBank speech data ($N = 291$) obtained during picture descriptions in NPT. We validated model generalisability through an independent pilot study with data collected out-of-clinic ($N = 22$) including from non-English speakers. We considered several ML models and extracted various acoustic and linguistic features using conventional methods based on domain knowledge and transformer-based pre-trained language models. Given our study's focus on clinical applicability, we prioritised interpretable features to inform clinicians of changes in spoken language patterns that indicate cognitive decline. Our final model incorporates 100 NLP-based features—including lexical, semantic, and psycholinguistic features—extracted from individual transcripts obtained through ASR.

The best-performing RF-NLP model for ADRD detection achieves a ROC-AUC of 85.7 (95% CI = 83.8–87.6) on the test set. This performance is comparable to previous studies using the same DementiaBank dataset[20,25,44], and outperforms results from models trained on interpretable features[45]. While previous studies reported performance from a single run on the test set, our experiments used 10 bootstrap repeats to ensure superior reproducibility. In addition, previous studies reported accuracy as the main performance metric, whereas we prioritised ROC-AUC since it is based on predicted probability scores (instead of discrete class labels) and measures the ability to classify true ADRD cases while minimising false positives.

On the independent pilot dataset, the model achieves a lower ROC-AUC of 65.4 (95% CI = 54.9–70.1), though the sensitivity is maintained (70% vs. 69.4% in the DementiaBank test set). It is important to note that the main clinical utility of using speech and language biomarkers lies in early screening, making sensitivity an important metric for correctly identifying individuals at risk. However, other steps should be taken to minimise the burden and effect of false positives. For example, structured verbal tasks (e.g. describing a picture) could be used as a second-tier screening to improve the specificity of the model after initial analysis of longitudinal conversational speech. Furthermore, most participants on the pilot dataset had MMSE scores corresponding to the MCI and CN groups (mean MMSE = 24.9, see Table 1). The higher proportion of participants with MCI also likely contributed to model uncertainty. Individuals in the MCI group often exhibit less pronounced changes in language[40], making it more difficult for the model to distinguish between cognitive groups.

The RFR achieves a MAE of 3.7 (95% CI = 3.7–3.8) on MMSE prediction, outperforming previous results using the same DementiaBank

dataset to predict MMSE as a continuous outcome[20,25]. The model then achieves a comparable MAE of 3.3 (95% CI = 3.1–3.5) on the pilot dataset, demonstrating its generalisability to speech data collected in real-world settings outside clinics. This result is particularly informative given that the model is trained only on the DementiaBank dataset, and was exclusively used for predicting MMSE on the pilot dataset. This suggests our model captured transferable knowledge and could operate as an out-of-the-box solution without requiring re-training.

In conjunction with our clinical team (including neurologists and psychiatrists), we introduce a traffic-light stratification system that categorises model predictions into clinically actionable risk scores. This approach results in a higher mean ROC-AUC on the test set, particularly with a 13% increase in specificity on the test set. Similarly, on the pilot dataset, the analysis yields a higher ROC-AUC and a 21% increase in specificity at the expense of reduced sensitivity. Although the small size of the training dataset limits broader conclusions on clinical effectiveness, this risk stratification analysis offers a comprehensive approach to actionable triage, which has not been explored previously in the context of ADRD screening from spoken language. Future studies with larger cohorts could use this approach to alert clinicians to individuals with increased risk and to optimise medical resource allocation, ultimately enabling more personalised and timely interventions.

Feature importance analysis enhances interpretability of our ML pipeline by identifying the linguistic features most predictive of ADRD. Our findings reveal that increased reliance on pronouns, particularly impersonal pronouns (e.g., 'that', 'it', 'this'), greater linguistic disfluency, and lower lexical diversity with repetitive language all contribute to a higher probability of ADRD, consistent with previous literature[46–49]. The frequent use of pronouns and high-frequency words likely indicates empty, vague, or non-specific speech, a known characteristic of cognitive decline[50]. We also found that language associated with reduced analytical thinking, decreased use of words reflecting a psychological state of completion (e.g., 'enough', 'full', 'complete'), and higher use of adverbs (e.g., 'there', 'so', 'just') all contribute to positive predictions. These findings suggest that participants with ADRD exhibit a decline in words reflecting cognitive processes related to structured and logical thinking. Words related to psychological completion were less common, potentially reflecting difficulties in articulating complete thoughts. Additionally, longer sentences and more frequent references to family were associated with a lower probability of ADRD (i.e., true negative cases). These findings suggest that CN participants tend to provide more

detailed and contextually rich descriptions of the Cookie Theft picture with greater inferences regarding relationships. Further investigation by language and cognition experts is needed to generalise these findings.

The proposed ML pipeline can be applied for feature adaptation or reduction to enhance explainability and clinical applicability. Future work could therefore explore acoustic disfluencies such as pauses, hesitations or false starts and combine them with the most predictive linguistic features identified in this study. Furthermore, the use of this picture as a stimulus has been subject to criticism due to traditional gender stereotypes and limited representation of racial and ethnic diversity[51]. Updated and culturally sensitive versions of the picture and the inclusion of conversational speech beyond structured tasks warrant future investigation.

We acknowledge limitations in our study that point toward future research directions. The use of ASR systems, such as OpenAI's Whisper[31], introduces transcription errors, particularly for participants with severe cognitive impairment and higher speech disfluency (see ASR robustness across the different cognitive groups in Supplementary Fig. 6). While ASR can affect the extraction of linguistic features, its use was an intentional design choice to assess the feasibility of automated screening of cognitive impairment in real-world settings, where human annotation is impractical. Prior work on speech from patients with stroke has shown that fine-tuning Whisper on pathological speech can improve recognition accuracy[52], which we highlight as a promising future direction. The quality of the DementiaBank audio data used for training can also impact the accuracy of the linguistic and acoustic features extracted for model development. Manual transcriptions on DementiaBank data showed a high correlation between linguistic features derived from participant-only transcriptions and those that included short segments of administrator speech, supporting our decision to proceed without automatic speaker diarisation. As these techniques improve, incorporating them as a pre-processing step could further enhance future analysis.

Cognitive screening tools developed primarily in white, English-speaking populations may not generalise well to more diverse populations due to differences in education and cultural backgrounds[53]. While our pilot study included a subset of speech recordings in Spanish, to improve the generalisability of our findings, future studies should include larger, more culturally diverse populations and explore predictive features in languages beyond English. Furthermore, automated translation was used for linguistic feature compatibility with the trained model, ensuring an end-to-end pipeline. This approach provides a realistic validation of model performance on an independent test set across a different demographic, language, and setting, without re-training or fine-tuning. Similar cross-linguistic approaches have been used in prior work[54,55], though this carries the risk of losing lexical nuance. Future research could investigate language-agnostic features and transfer learning to support multilingual analysis. A multilingual sensitivity analysis on the pilot dataset collected in this study is included in Supplementary Table 8.

Future studies could integrate other predictive features, such as age, sex, education, and family history of dementia, which could improve models for ADRD screening and severity prediction. Integrating additional in-home behavioural data, comorbidities, and individual health events such as hospitalisations or infections could further improve prediction performance and enhance clinical applicability[56,57]. Collecting longitudinal data from participants would also be valuable for predicting disease progression over time. Moreover, moving beyond binary classification would broaden the use of our methods, and future studies could include MCI or other neurodegenerative disorders as prediction classes.

## Conclusions

Our study offers a clinically oriented, comprehensive pipeline that integrates: 1) automated screening of cognitive impairment from spoken language with external validation of our parameterised model unchanged to an independent dataset collected out-of-clinic including non-English speakers, 2) a risk stratification approach for clinically actionable triage, 3) fairness analysis across demographic subgroups alongside calibration of predicted risk with severity of cognitive impairment, and 4) predicted state of cognitive decline to support future longitudinal studies and prognostic assessment. Our interpretable predictive modelling approach can be integrated with home-based conversational AI. With consistent use, these technologies hold potential to become accessible and personalised tools that track trajectories of cognitive decline and alert clinicians to higher-risk individuals who require further diagnostic workup.

## Data availability
Source data for Fig. 1 and Fig. 2 are provided in Supplementary Data 1. Source data for Fig. 3 are provided in Supplementary Data 2. Source data for Fig. 4 are provided in Supplementary Data 3. The ADReSSo data and Lu corpus are available via DementiaBank[29]. The pilot data used in this study may be available by the corresponding author upon reasonable request. Requests regarding data can be directed to M.R.L., R.V. or P.B.

## Code availability
The code used in this study, including scripts for the machine learning pipeline, is available at https://github.com/mariarlima/ml-speech-biomarkers and archived on Zenodo[58]. Please contact M.R.L for further enquiries.

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

## Acknowledgements

This work was supported by the UK Dementia Research Institute (award numbers UK DRI-7003 and UK DRI-7005) through UK DRI Ltd, principally funded by the Medical Research Council, and additional funding partner Alzheimer's Society. This work was also funded by the Research England Grand Challenge Research Fund (GCRF) through Imperial College London and Imperial College London's President's PhD Scholarships. Infrastructure support for this research was provided by the NIHR Imperial Biomedical Research Centre (BRC) and the UKRI Medical Research Council (MRC). PB is funded by the Great Ormond Street Hospital and the Royal Academy of Engineering, and EPSRC/NIHR (grant number EP/W031892/1). FG is funded by MRC (grant number MR/T001402/1).

## Author contributions

M.R.L.: Conceptualisation, Model Development & Evaluation, Formal Analysis, Investigation, Methodology, Writing-Original Draft, Review and Editing; A.C.: Formal Analysis, Visualisation, Writing-Review and Editing; F.G., R.N., M.M.: Conceptualisation, Methodology, Writing-Review and Editing; R.V., P.B.: Conceptualisation, Methodology, Supervision, Funding Acquisition, Writing-Original Draft, Review and Editing,

## Competing interests

The authors declare no competing interests.
