## [Transparent Peer Review file · Communications Medicine]

Evaluating Spoken Language as a Biomarker for Automated Screening of Cognitive Impairment

Corresponding Author: Dr Maria Lima

Version 0:

Reviewer comments:

Reviewer #1

(Remarks to the Author)

Overall:

There is a lot to like about this paper; it is well-written, in a clear and explanatory manner. Nonetheless, I outlined several points for clarification and improvement below.

Introduction:

Flows well, no comments.

Methods:

- What is an 'ADRD diagnosis', is it a diagnosis of all-cause dementia or a diagnosis of Alzheimer's disease or another specific dementia syndrome? Please specify. Given the name of the dataset (Alzheimer's Dementia Recognition through Spontaneous Speech only) do the individuals specifically have Alzheimer's disease?

- Terminology: Initially, it seemed like the terms 'linguistic' 'lexical' and 'acoustic' were sort of used interchangeably in the manuscript—only after digging through the supplementary data and reading further into the methods did it become clearer that the initial 'linguistic and acoustic features' were reduced to just linguistic features, and that the linguistic features were mainly labeled as lexical features (or lexical-semantic features). The manuscript could be clearer at several points. For example,

Line 118 talks about both lexical features and linguistic features, but it turns out the authors are referring to the same set of features, so why use two different terms? Line 172: this paragraph is not just about linguistic features, header should be 'linguistic and acoustic features'. Line 193: lexical-based features are features only related to words – do you mean linguistic-based features (from transcripts)? And does Line 289 'lexical-based features' really only mean lexical features? I would argue you move slightly broader into linguistic features (number of words per sentence arguably goes into syntactic features). Readers who are not well-versed into linguistics may get lost in this different terminology, so the authors may want to clarify either by including definitions of linguistic vs lexical vs semantic features, or use mainly the word 'linguistic features' where it says 'lexical' now, and clarify elsewhere (methods or discussion) that the majority of the linguistic features are lexical-semantic features. Also: Line 57: digital voice biomarkers are different than speech & language markers—voice markers includes jitter/shimmer and arguably several acoustic measures, but not language markers; if anything, the field is leaning more to calling it 'digital speech biomarkers'.

- As a reader I have a lot of questions about the extracted features and their operational definitions, even after reading the table in the supplementary materials. For example, how is the feature of 'analytical thinking' developed (what goes into it—is that LIWC's secret or can you report on it)? What is the reasoning behind 'fulfill words' having higher semantic complexity? How did you measure disfluencies (line 366), acoustically or linguistically, and which disfluencies exactly (pauses filled and/or silent, hesitations, false starts, fillers, etc etc)? Which package was used to extract idea density (e.g., the tool 'CPIDR' is not great for idea density in AD, see <https://aclanthology.org/K17-1033.pdf> so it's helpful to know which tool was used). What are 'overall linguistic variables' in line 383? Also, supplementary table 5 labels all LIWC variables as 'semantic psycholinguistics' which is not true, e.g., word count, words per sentence, number of pronouns, nouns, verbs etc are lexical features, not semantic. And some of the 'lexical diversity' measures can be marked as semantic measures (such as idea density).

- Line 182: please provide details in-text on which Whisper model was used (e.g., medium, large, turbo, and version number)—the supplemental materials mention 'medium' but do not mention version (both should be mentioned in the manuscript). Additionally: the medium model is not optimal (creates quite some mistakes such as hallucinations or incorrect words, which

increase with impaired speech), and the discussion mentions not doing manual checks was an intentional choice and acknowledges that it's particularly more erroneous in cognitively impaired speech, so why not choose a better performing Whisper model (large or turbo)?

- Why translate Spanish transcripts to English? That is not appropriate for linguistic analysis, especially not for lexical features. If the authors want to include analysis on Spanish data, there are other ways, e.g., https://www.isca-archive.org/interspeech_2022/pereztoro22_interspeech.pdf
- The described methods for ML models seem solid, no changes requested.
- The use of more family-related words seems to be a direct characteristic of the cookie theft picture (which depicts a family), most likely not transferable to other picture description tasks, so the authors should be careful to include such a stimulus-bound variable in their models. Related: It would be good to acknowledge the controversy around the cookie theft picture in the discussion—not the fault of the authors, but just to make readers aware and hopefully move the field away from using that picture in future studies (doi:10.1001/jamaneurol.2022.1409). Even more so reason to rethink the inclusion of stimulus-bound features in a model, if the authors intend for the model to be transferable to other speech tasks (and this particular task is being faded out).

Results:

- Line 321: The authors should be careful in their wording; they are not predicting 'risk of dementia' in this sample, the model predicts probability / classification based on people having dementia or not having dementia; the field uses 'dementia risk' or 'risk of dementia' to mean the likelihood someone WILL develop dementia (in the future), which requires longitudinal data. The paragraph headers in line 254 and 331 and subsequent wording of 'risk' in the manuscript should also be adjusted. For example, line 382 is not '91% risk' but '91% probability'
- The selected model performs well in the test and validation sets from the DementiaBank, not in the pilot study—besides being a very small sample, the pilot study data are also weaker because of how Spanish data are treated (i.e., translation). I don't think the Spanish data contribute to 'generalizability' of the findings, as it is unclear whether the Spanish data perform as well as the English data (the pilot is a mix of the two languages)—perhaps the model doesn't perform well because of the inclusion of Spanish-automatically-translated-to-English data? Can the authors do a sensitivity analysis on the English-only and Spanish-only data from their pilot?
- 'Real World' is not the correct term to use for the pilot study conducted in this paper; real world data means to data collected outside of a controlled setting, meanwhile this data is still cookie theft data collected in a controlled setting from a selected volunteer-based sample. Real-world data would be speech "in the wild", not task-related. This term should be removed from the full manuscript. This pilot is just another small dataset to validate the model developed on the test set; the difference is that it's not from the DementiaBank, it's not 'real-world'.

Discussion:

- Generally well-written, see some comments above about points to be discussed in the Discussion.
- Additionally, can the authors discuss the diversity of the sample and how it reflects the population this sample was taken from (i.e., the general population of the UK)? That will speak to the extent of generalizability of this sample—historically, a lot of research has been performed in high-educated White samples, acknowledging the composition of this sample highlights either that future research must be more inclusive, or highlights that the DementiaBank is a diverse source which increases the generalizability of the findings. Does the DementiaBank have any data on education and race/ethnicity of the included individuals (if yes, please also include in Table 1)?

Minor:

- Line 135/136: moderate and severe are missing nouns—e.g., moderate dementia and severe dementia. Or is it moderate Alzheimer's disease and severe Alzheimer's disease?
- Line 158: a = an (an ADRD diagnosis)
- Line 377: 'this' needs to be followed by a noun, otherwise antecedent remains ambiguous

Reviewer #2

(Remarks to the Author)

This study presents an interpretable machine learning approach for the screening of cognitive impairment using language. The authors leverage linguistic features from Addresso (DementiaBank) speech recordings to develop models for Alzheimer's Disease and Related Dementias classification. They validate their models on the Lu corpus (also from DementiaBank) and a real-world dataset, demonstrating potential for generalizability beyond controlled datasets. The work contributes to the growing field of digital biomarkers and aligns well with Communications Medicine's mission of publishing research with high clinical impact.

The authors found that machine learning models trained on linguistic features achieve strong classification performance, with a Random Forest model using lexical features achieving an area under the curve (AUC) of 85.7% on a test dataset. The inclusion of a pilot dataset collected from older adults supports external validity, though the sample size is relatively small (N=22).

This study improves upon prior work by emphasizing explainability, real-world validation, and risk stratification, which are crucial for clinical adoption. The use of SHAP analysis to interpret feature importance and cross-validation with bootstrapping for reproducibility further strengthens the study's impact.

Regarding reproducibility, the authors provide:

- Details on data sources (DementiaBank, pilot study), with clear inclusion criteria.
- Code and dataset availability statements, though explicit dataset pre-processing steps could be more detailed.
- Feature selection and model training descriptions, enabling replication by other researchers.

Major concerns:

The study relies on OpenAI's Whisper for automatic speech transcription. While the authors acknowledge transcription errors, particularly in participants with severe impairment, further analysis on how ASR errors impact model performance would be beneficial. Is there a relationship between amount of ASR error and prediction of AD? Exploring ASR robustness across different cognitive states could further strengthen the analysis. Additionally, while the pilot study includes Spanish speakers, the main models are trained on English data. This could be problematic. Is there work that the authors could point readers to for validity within this sort of multilingual scenario?

Further information would be helpful for the second paragraph of the Predictive Modeling section. How did the authors split the data to do both 10 fold cross validation and have a held out test set from the Addresso dataset?

For the risk scores model, excluding uncertain predictions from accuracy reporting can be problematic because it artificially inflates performance metrics.

A 100-feature model is difficult to consider fully explainable, even if it uses interpretable linguistic features. While SHAP analysis helps identify important features, having 100 variables means that understanding individual predictions remains complex, especially for clinicians who need transparent decision-making tools.

I would urge the authors to make a stronger case for the novelty of this work. As noted, there are many studies that come before them that apply NLP methods to the cookie theft task - both feature based and LLM based.

Version 1:

Reviewer comments:

Reviewer #2

(Remarks to the Author)

I appreciate the authors' efforts to revise the manuscript and address prior feedback. The revised version is clearer and better organized, and I commend the team for improvements in transparency and reproducibility. However, despite these efforts, I remain unconvinced that this work offers sufficient novelty to warrant publication.

The central concern is that the paper's primary contributions - using explainable machine learning on linguistic features extracted from speech to detect cognitive impairment and stratify AD risk closely mirror what has already been accomplished in prior work. For example, Chandler et al. (2023) [<https://alz-journals.onlinelibrary.wiley.com/doi/epdf/10.1002/dad2.12516>] presents a comparable pipeline: remote speech-based assessment using acoustic and linguistic features, explainable modeling, and attention to clinical translatability through a dashboard and risk interpretability. Lindsay et al. (2021) [<https://www.frontiersin.org/journals/aging-neuroscience/articles/10.3389/fnagi.2021.642033/full>] investigated generalizable language features in AD detection from the AdreSSo dataset to a multilingual setting. The parallels between the current work and previous published work raise concerns about the novelty of the present submission. The authors' claim of novelty in generalizability, interpretability, and use of in-home data is not substantially distinct from what others have demonstrated.

While the study is competently executed and relevant to the field of digital biomarkers for dementia, in my opinion, it does not meet the bar for novel contribution in its current form. I recommend rejection unless the authors can more clearly differentiate their methodological contributions or demonstrate unique findings not previously shown.

Reviewer #3

(Remarks to the Author)

The authors have made a clear and thoughtful effort to address the reviewers' concerns, resulting in a substantially improved and more coherent manuscript. Analyzing the additional pilot dataset to evaluate the generalizability of the Dementia-Bank-derived model represents a valuable and innovative step forward. Importantly, the limitations of this additional dataset are transparently acknowledged, which strengthens the overall credibility of the work.

The authors note an important result: binary classification of AD versus cognitively normal (CN) participants typically yields stronger performance than the more challenging MCI versus CN task. However, it is precisely the latter classification problem that poses the greatest clinical challenge and where natural language processing, acoustic features, and machine learning approaches can offer the greatest translational value.

To further situate this contribution within the existing literature, the manuscript could benefit from citing prior studies that have applied interpretable machine learning models to the detection of cognitive decline, particularly those addressing the more difficult three-way classification (AD, MCI, CN). Doing so would not only highlight the novelty of the present work but also contextualize its potential impact in advancing clinically meaningful applications.

eg., <https://ieeexplore.ieee.org/abstract/document/9769980>

<https://arxiv.org/pdf/2506.11119>

<https://alz-journals.onlinelibrary.wiley.com/doi/full/10.1002/dad2.12516>

Similarly, the authors might consider citing accuracy results from the top teams in ADReSSo Challenge for comparative purposes.

Version 2:

Reviewer comments:

Reviewer #2

(Remarks to the Author)

Thank you for clarifying the novelty of your study in context of previously cited works. The additions and revisions to the manuscript strengthen its contributions. I recommend the article for acceptance to Communications Medicine.

Reviewer #3

(Remarks to the Author)

I am satisfied that the authors have addressed the reviewers' concerns in this revision and appreciate their efforts to emphasize the added value of their work in relation to prior studies.

Reviewer Response

We thank the reviewers for their thoughtful and constructive feedback. We appreciate that R1 highlighted the clarity of our manuscript and solid description of the machine learning approach. We are encouraged by R2's recognition of our study's contribution to the growing field of digital biomarkers, its alignment with the mission of Communications Medicine to publish research with high clinical impact, and its distinction from prior work on validation using speech data collected in-residence, explainability, and risk stratification as key strengths for clinical adoption. We are also grateful for R2's acknowledgement of transparency and reproducibility of our research. The valuable feedback guided our improvements, highlighted in the revised manuscript in blue. A detailed response to all reviewer comments, quoted in *italics*, follows.

Additionally, we added the *TRIPOD + AI statement* (Version: 11-January-2024) following guidance for reporting clinical prediction models (see *Supplementary Material 15*).

Reviewer 1

1.1. *What is an 'ADRD diagnosis', is it a diagnosis of all-cause dementia or a diagnosis of Alzheimer's disease or another specific dementia syndrome? Please specify. Given the name of the dataset (Alzheimer's Dementia Recognition through Spontaneous Speech only) do the individuals specifically have Alzheimer's disease?*

We thank the reviewer for this important clarification request. In the DementiaBank ADReSSo dataset the Alzheimer's disease diagnosis was based on clinical diagnosis of patients referred from memory clinic largely based on Diagnostic and Statistical Manual of Mental Disorders, 3rd Edition criteria [1]. Given the absence of fluid or amyloid imaging biomarkers, it would be expected that up to 20% of the data may be misclassified [2]. Therefore, we opted for a more inclusive terminology of Alzheimer's Disease and Related Dementias, 'ADRD'. We clarified this in the revised manuscript; Section *Speech Datasets* now states (line 162): "ADRD diagnosis here describes a clinical diagnosis of Alzheimer's dementia in the absence of fluid and imaging biomarker confirmation. Therefore, the label is likely to include all-cause dementias".

1.2. *Terminology (...) the manuscript could be clearer at several points (...).*

We thank the reviewer for their meticulous assessment and apologise for the lack of clarity. We clarified the terminology used, ensuring consistency throughout the manuscript by using linguistic features as the overarching term for features derived from transcripts. We clarified that the majority of linguistic features used in the final model are lexical-semantic (see *Methods – Acoustic and Linguistic Features*). The reviewer correctly noted that both linguistic and acoustic features were explored for comparison of model performance on the validation set (with results included in Appendix Table 7). We selected the best-performing model using linguistic features for its predictive performance, interpretability and clinical utility. We have clarified this rationale in *Methods – Model and Feature Selection* (line 270-277). We also agree that speech biomarkers is a more appropriate term in this context and have revised the manuscript accordingly.

1.3. *How is the feature of ‘analytical thinking’ developed (what goes into it—is that LIWC’s secret or can you report on it)?*

The analytical thinking feature is a factor-analytically derived metric intended to capture logical, formal, and hierarchical thinking. It is based on the relative use of function words such as articles, prepositions, pronouns, and auxiliary verbs. Using large text corpora, LIWC developers applied factor analysis to identify patterns that correlate with more analytical writing styles. For new texts, the score is computed by weighting function word frequencies by their factor loadings and standardising the result on a 0–100 scale, placing each text on a narrative-to-analytical continuum, as used in previous psychology studies [3-5]. We thank the reviewer for this clarification request and added it to *Supplementary Material 5*.

1.4. *What is the reasoning behind ‘fulfill words’ having higher semantic complexity?*

We agree with the reviewer that while the “fulfill words” feature may provide insight into psychological states conveyed in spoken language (sense of fulfilment and satisfaction with the response), as proposed by LIWC, it should not be used in isolation as an indicator of semantic complexity. On further reflection we have amended this sentence and removed the association with semantics.

1.5. *How did you measure disfluencies (line 366), acoustically or linguistically, and which disfluencies exactly (pauses filled and/or silent, hesitations, false starts, fillers, etc etc)?*

We agree with the reviewer that this term required clarification. In our analysis, disfluency is a normalised linguistic feature that aggregates the frequency of the following LIWC-defined categories: ‘nonfluencies’ (e.g., *uh, um, oh*), ‘fillers’ (e.g., *you know, I mean, like*), and ‘assent words’ (e.g., *uh-huh, yeah, ok*). These sub-features are also considered individually in the analysis. We have now renamed it to ‘linguistic disfluency’ and revised Section *Linguistic Feature Importance* to explicitly define this feature (line 387-396). As shown in the SHAP plot in Figure 3, linguistic disfluency was among the top 12 predictive features in the Random Forest model, with increased use of assent words as an individual feature also associated with higher probability of ADRD.

We note that acoustic features derived from eGeMAPS and pre-trained deep models (e.g., *data2vec*), which could have captured acoustic elements of dysfluency (such as sound repetitions, sound elongations or hesitations), were also explored for comparison in the validation set (Table 7). We found that linguistic features outperformed acoustic-only and fusion models. Given our goal of a scalable, end-to-end screening pipeline for use in real-world settings beyond clinics, we prioritised features that do not require manual annotation. However, future work could explore acoustic disfluencies such as pauses, hesitations or false starts in greater depth, which we now note in *Discussion* (lines 566-570).

1.6. *Which package was used to extract idea density (e.g., the tool ‘CPIDR’ is not great for idea density in AD, see <https://aclanthology.org/K17-1033.pdf> so it’s helpful to know which tool was used)*

We thank the reviewer for highlighting the limitations of CPIDR, as discussed in Sirts et al. (2017). We implemented a manual calculation of propositional idea density (PID) using part-of-speech (POS) tags derived from NLTK Python toolkit; we consider the number of expressed propositions (i.e. distinct facts or notions contained in a text with verbs, adjectives, adverbs, prepositions, and conjunctions) divided by the total number of words as in prior work (e.g., [6]). We clarified this in *Supplementary Table 5*. While we were not aware of the DEP/DEP-R approach that incorporates dependency structures and idea repetition in AD transcribed speech, our primary contribution lies in the development of a scalable machine learning (ML) approach for automated home-based screening and risk assessment with a focus on interpretability and clinical utility. We believe the current approach provides a valid proxy within this context. Future work could extend and refine the linguistic feature set, including dependency-based propositional density or embedding-based semantic idea density (SID), within the framework proposed.

1.7. *What are ‘overall linguistic variables’ in line 383?*

We agree with the reviewer that this term required clarification. The overall linguistic variable, as defined in LIWC-22 Psychometrics Manual [7], is a normalised feature that aggregates the frequency of LIWC-defined linguistic dimensions, including function words, determiners, prepositions, auxiliary and common verbs, adverbs, conjunctions, negations, adjectives and quantities. These sub-features are also considered individually in the analysis. We have revised the manuscript to refer readers to *Supplementary Material 5 (line 413)*, where this feature is now explicitly defined.

1.8. *Also, supplementary table 5 labels all LIWC variables as ‘semantic psycholinguistics’ which is not true, e.g., word count, words per sentence, number of pronouns, nouns, verbs etc are lexical features, not semantic. And some of the ‘lexical diversity’ measures can be marked as semantic measures (such as idea density).*

The reviewer is correct that the original labels lacked precision in terminology. We corrected these in *Supplementary Table 5* to “Lexical diversity and semantic complexity” and “Lexical and semantic psycholinguistic” features. We also revised the manuscript for consistency.

1.9. *Line 182: please provide details in-text on which Whisper model was used (...). Additionally: the medium model is not optimal (creates quite some mistakes such as hallucinations or incorrect words, which increase with impaired speech), and the discussion mentions not doing manual checks was an intentional choice and acknowledges that it’s particularly more erroneous in cognitively impaired speech, so why not choose a better performing Whisper model (large or turbo)?*

We used OpenAI’s Whisper medium.en model (version 20240930), now specified (*line 201*) and *Supplementary Material 5*. We acknowledge the limitations of automatic speech recognition (ASR) systems more broadly, particularly in cognitively impaired speech, and address this in the Discussion (*line 594*). To further assess transcription quality, we analysed a representative subset of 61% of DementiaBank data – which covers at least 50% of speech samples per

cognitive group (cognitively normal, mild, moderate, and advanced). We computed Word Error Rate (WER) using the metric from HuggingFace (<https://huggingface.co/spaces/evaluate-metric/wer>). As we now report in *Supplementary Material 7*, there was a WER of 15.7% (SD = 19.8%), which aligns with recent reports of Whisper's performance on Alzheimer's and control speech, with mean WERs of 19.3% (Whisper medium.en), 31.9% (Whisper large-v2) [8], and 30.2% (Whisper large) [9]. Future iterations of this work could explore fine-tuning Whisper on annotated pathological speech datasets. Prior work by our group [10] has shown that such fine-tuning improves recognition accuracy on patients with stroke related pathological speech. We now include this as a promising future direction in Discussion (line 604).

The choice of the medium model was guided by a balance between transcription accuracy, computational efficiency, and scalability in real-world and low-resource clinical settings. We note that our primary contribution lies in the development of a scalable ML approach for automated screening and risk assessment with enhanced interpretability and clinical applicability, rather than to optimise transcription performance per se. We see this study as a framework that can integrate newer ASR models as they become available in this rapidly evolving field, such as CrisperWhisper [11], which can capture disfluencies and fillers more reliably.

1.10. *Why translate Spanish transcripts to English? That is not appropriate for linguistic analysis, especially not for lexical features. If the authors want to include analysis on Spanish data, there are other ways e.g.,*

https://www.isca-archive.org/interspeech_2022/pereztoro22_interspeech.pdf

We thank the reviewer for raising this important point. We agree that translating Spanish transcripts to English risks losing lexical nuance. Our rationale was to develop and evaluate an end-to-end ML pipeline with clinical applicability in real-world, multilingual settings. The best-performing model was trained on English linguistic features. Due to the limited size of the Spanish pilot subset (N=8), training or calibrating a separate model would have overfit. To generate compatible input features, we used automated translation, reviewed by a fluent Spanish speaker, and applied the model to the Spanish data only as an independent test. This approach allowed us to give a real reflection of how the model performs, on an independent test set, across a new cohort, language, and setting, without over-engineering features.

Performance results on the pilot dataset showed that the model can still perform in an independent test set from a different multi-lingual population. However, we agree that future work should explore extracting language-agnostic features or learning shared representations across languages (e.g., Pérez-Toro et al., 2022) to support cross-linguistic analysis without automated translation. We have revised the *Discussion* to reflect this point (lines 620-634) and include a follow-up sensitivity analysis between different languages if large enough samples are available. We provide more details in our response to a subsequent comment by the reviewer (see 1.13).

1.11. *The use of more family-related words seems to be a direct characteristic of the cookie theft picture (which depicts a family), most likely not transferable to other picture description tasks, so the authors should be careful to include such a stimulus-bound*

variable in their models. Related: It would be good to acknowledge the controversy around the cookie theft picture in the discussion (...).

We acknowledge that the use of family-related words can be stimulus-bound given that the Cookie Theft picture depicts a domestic scene. We believe this feature is still of interest in capturing the contextual relevance of spontaneous speech produced by the target population, so we chose to keep it in our 100-dimensional linguistic feature set. Similar features could be engineered for different pictures by grouping context-relevant words linked to the content of each picture or more broadly, the conversational task. Furthermore, we chose the Cookie Theft picture due to its standardisation and the availability of open-access speech data from DementiaBank, which facilitated model development, validation, and benchmarking. However, we recognise the broader limitations and controversy surrounding this stimulus, including its cultural specificity and ecological validity, and we now acknowledge this in the *Discussion* (line 570).

1.12. *Line 321: The authors should be careful in their wording; they are not predicting ‘risk of dementia’ in this sample, the model predicts probability / classification based on people having dementia or not having dementia; the field uses ‘dementia risk’ or ‘risk of dementia’ to mean the likelihood someone WILL develop dementia (in the future), which requires longitudinal data. The paragraph headers in line 254 and 331 and subsequent wording of ‘risk’ in the manuscript should also be adjusted. For example, line 382 is not ‘91% risk’ but ‘91% probability’.*

We agree that the term ‘risk’ and use of the proposed risk stratification approach in the context of screening cognitive impairment required clarification. We note that we refer to *risk of dementia*, we refer specifically to spoken language-based indicators of risk. This does not encompass other biological or physiological aspects of dementia risk. In consultation with our clinical experts, we implemented a post hoc risk stratification to enhance the clinical applicability of model predictions and reflect how such an automated screening tool could be used in clinical practice to support clinicians in triaging higher-risk individuals who may benefit from further diagnostic workup. To this end, we stratified prediction scores from the best-performing model into three ADRD risk groups: Green (low risk), Amber (medium risk), and Red (high risk). Similar risk stratification methods have been validated in urinary tract infection detection and agitation monitoring in dementia care [12]. This approach does not indicate longitudinal risk prediction but provides an interpretable framework to reflect how clinicians might interpret uncertainty in model outputs. Such stratification is intended to reduce false positives, streamline triaging, and inform translational pathways for clinical adoption. We revised the manuscript to refer to model outputs as predicted probabilities as correctly pointed out by the reviewer, updated the title *Stratification of Risk Scores for Clinical Adoption* (line 278) and retain the term risk when referring to the stratification approach.

1.13. *The selected model performs well in the test and validation sets from the DementiaBank, not in the pilot study—besides being a very small sample, the pilot study data are also weaker because of how Spanish data are treated (i.e., translation). I don’t think the Spanish data contribute to ‘generalizability’ of the findings, as it is unclear whether the*

Spanish data perform as well as the English data (the pilot is a mix of the two languages)—perhaps the model doesn't perform well because of the inclusion of Spanish-automatically-translated-to-English data? Can the authors do a sensitivity analysis on the English-only and Spanish-only data from their pilot?

We thank the reviewer for this valuable observation. On this occasion, we did not train the model with pilot study data; rather, we used the model trained on DementiaBank and validated its performance on the pilot dataset. This pilot dataset, while small in size, was included to validate the pre-trained model in a different demographic and in-residence setting, using the same standardised speech task. To the best of our knowledge, no prior studies have evaluated ML models for automated screening of cognitive impairment across demographics, settings, and languages. Additionally, the model was neither trained nor fine-tuned on this dataset, making it a fully independent test, consistent with ML best practices.

In response to the reviewer's suggestion, we conducted a sensitivity analysis by splitting the pilot data into English-only (N=14) and Spanish-only (N=8) subsets. We computed classification metrics for each subgroup using 10 bootstrap repeats and report 95% confidence intervals (see the newly added *Supplementary Material 14 "Pilot Study: Multilingual Sensitivity Analysis", Table 11*). Not surprisingly, the model performance drops for the Spanish-only subset. While sensitivity remained comparable between English (71.0%, 95% CI: 61.8–80.2) and Spanish samples (67.5%, 95% CI: 52.8–82.2), specificity decreased in the Spanish group, leading to a drop in overall accuracy to 53.8% (95% CI: 47.7–59.8).

This drop likely reflects two factors: (1) lexical noise introduced by automated translation, as discussed in a previous response (see 1.10); (2) the higher average MMSE in the Spanish subgroup of 26.1 (3.6) vs. 24.2 (3.9), suggesting that many participants were near the cutoff (MMSE = 26) where model uncertainty is higher, as we demonstrated in Figure 1. We also note differences in the distribution of cognitive impairment groups across subsets. The English subset included participants in the moderate group – where the model is more confident – whereas the Spanish subset only included individuals in the CN or MCI groups. We anticipate that performance on Spanish-only samples would improve with a larger cohort that includes individuals with more advanced cognitive decline. Although the limited number of Spanish participants prevents broader conclusions, we argue the model does not lose its overall utility.

1.14. *'Real World' is not the correct term to use for the pilot study conducted in this paper (...). This term should be removed from the full manuscript. This pilot is just another small dataset to validate the model developed on the test set; the difference is that it's not from the DementiaBank, it's not 'real-world'.*

Our use of the term “real-world” was intended to distinguish data collected in-residence, that is, outside clinics. Although we believe there is merit in collecting speech from target population in residential settings, the reviewer is right that this was still controlled using a standardised protocol and not deployed for unsupervised data collection over time. We have revised the manuscript to refer to the pilot dataset as “speech data collected in-residence from older adults”.

1.15. *Additionally, can the authors discuss the diversity of the sample and how it reflects the population this sample was taken from (i.e., the general population of the UK)? That will speak to the extent of generalizability of this sample (...). Does the DementiaBank have any data on education and race/ethnicity of the included individuals (if yes, please also include in Table 1)?*

The DementiaBank dataset does not include detailed demographic information such as education level or race/ethnicity, and therefore we are unable to report these characteristics in Table 1. For the pilot dataset, participants self-reported their races and ethnicities: 14 African-American, 7 White, 8 Hispanic Latino; 1 participant did not disclose, now included in Methods (line 184). We fully agree with the importance of including demographically diverse samples for developing equitable and generalisable cognitive screening tools. We note this limitation in the Discussion and highlight the need for future studies to incorporate more representative cohorts to support cross-cultural and multi-lingual applicability (line 617).

1.16. *Minor: Line 135/136: moderate and severe are missing nouns—e.g., moderate dementia and severe dementia. Or is it moderate Alzheimer’s disease and severe Alzheimer’s disease?; Line 158: a = an (an ADRD diagnosis); Line 377: ‘this’ needs to be followed by a noun, otherwise antecedent remains ambiguous.*

We thank the reviewer for their meticulous assessment. We corrected the minor typos and updated the manuscript accordingly.

Reviewer 2

2.1. *Regarding reproducibility, the authors provide: Code and dataset availability statements, though explicit dataset pre-processing steps could be more detailed.*

The DementiaBank audio recordings were acoustically pre-processed with noise reduction and volume normalisation (line 149). Following ASR transcription, linguistic feature extraction involved pre-processing steps such as lowercasing, punctuation removal, and tokenisation using the Python *spaCy* library. Our machine learning pipeline included hyperparameter optimisation using 10-fold cross-validation on the training set, with parameter ranges provided in *Supplementary Material 2*. We have revised the manuscript to more clearly describe these pre-processing steps to support reproducibility.

2.2. *The study relies on OpenAI’s Whisper for automatic speech transcription. While the authors acknowledge transcription errors, particularly in participants with severe impairment, further analysis on how ASR errors impact model performance would be beneficial. Is there a relationship between amount of ASR error and prediction of ADRD? Exploring ASR robustness across different cognitive states could further strengthen the analysis.*

We thank the reviewer for this valuable comment. We agree that the evaluation ASR robustness across different cognitive states strengthens the analysis. As noted in our response to Reviewer 1 (see 1.9), we analysed a representative subset comprising 61% of the DementiaBank training dataset, covering at least 50% of speech samples from each cognitive group. As expected, Word Error Rate (WER) increased with cognitive impairment severity. The highest error rates were observed in the severe impairment group (mean = 31.4%, SD = 12.8), while the cognitively normal (CN) group showed the lowest error rates (mean = 8.9%, SD = 9.1). These trends reflect known patterns of reduced fluency and increased disfluencies in advanced stages of dementia, which pose greater challenges for ASR systems. We added *Supplementary Material 7* (“Automatic Speech Recognition Robustness Across Cognitive Groups”) with Figure 8 to visualise WER distributions across cognitive groups.

2.3. *Additionally, while the pilot study includes Spanish speakers, the main models are trained on English data. This could be problematic. Is there work that the authors could point readers to for validity within this sort of multilingual scenario?*

We thank the reviewer for this observation and would like to reiterate our response to Reviewer 1 (1.10). The field of dementia detection has traditionally relied on single-language datasets, primarily in English, limiting cross-linguistic generalisability. However, recent work has begun to address this gap with cross-lingual approaches. Of particular relevance to our study, Perez-Toro et al. (2024) [13] conducted a between-language using classifiers trained on English-speaking AD patients and evaluated on Spanish data, Melistas et al. (2023) [14] evaluated zero-shot and few-shot setups by training on English and testing on Spanish and Greek data, and Shakeri et al. (2025) [15] examined multilingual datasets with automated translation, highlighting the need for language-specific strategies, as some languages benefit from multilingual training, while others achieve better performance with language-specific models. We revised the manuscript to contextualise our approach in *Discussion* (lines 620-634), We stress that this dataset was used solely as a pilot study for independent testing across a new cohort, language, and setting, without retraining or fine-tuning.

2.4. *Further information would be helpful for the second paragraph of the Predictive Modeling section. How did the authors split the data to do both 10-fold cross validation and have a held out test set from the Addresso dataset?*

The DementiaBank speech samples were used with a 70:30 train-test ratio balanced for demographics. The training subset was used for hyperparameter tuning via 10-fold cross-validation. The remaining 30% served as a held-out test set, not seen during any stage of model development. To estimate performance variability on the test set, we applied bootstrapping with 10 repeats. Each run used a bootstrap sample drawn from the training set to ensure robustness and reproducibility, rather than relying on a single evaluation. We have revised the manuscript (line 249-258) to clearly describe these steps.

2.5. *For the risk scores model, excluding uncertain predictions from accuracy reporting can be problematic because it artificially inflates performance metrics.*

The reviewer is right to note that using selective classification artificially inflates performance metrics. This risk stratification approach has been applied in remote monitoring and screening of UTIs in dementia care [12], as pointed out previously in 1.12. We note that thresholds for risk groups were defined on the validation set to mitigate the effects of overfitting and final performance was calculated on the test set only. The aim of the stratification was not to improve accuracy, but rather to reduce false alarms, enhance clinical interpretability, and reflect how such a tool could support clinicians. By categorizing patients into different risk groups, clinicians can prioritize those who require immediate attention and further diagnostic workup. By creating the amber (uncertain) group, we are reducing the number of False Positives and False Negatives. In a clinical pathway, this group can be re-assessed by other means beyond speech biomarkers compared to others with lower risk, therefore reflecting real-world clinical decision-making. We have clarified this rationale in the revised manuscript (line 368).

2.6. *A 100-feature model is difficult to consider fully explainable, even if it uses interpretable linguistic features. While SHAP analysis helps identify important features, having 100 variables means that understanding individual predictions remains complex, especially for clinicians who need transparent decision-making tools.*

To balance performance with explainability, we used SHAP values to identify both global and local feature contributions. While the overall SHAP summary plot highlights key predictors across the population, the framework also supports *individual-level* explanation, allowing clinicians to explore personalised predictions and the specific linguistic features contributing to a given risk score (such as those illustrated in Figure 10). Additionally, these features could further be grouped into categories. We acknowledge that future work can refine the feature set to further enhance interpretability. The proposed ML pipeline can be applied for feature reduction/adaptation. Moreover, we envision this tool to complement additional individual modalities such as in-home behavioural data, comorbidities, or hospitalisations to support early detection of cognitive decline and guide further clinical translational pathways. We have clarified this in the revised manuscript (line 567).

2.7. *I would urge the authors to make a stronger case for the novelty of this work. As noted, there are many studies that come before them that apply NLP methods to the cookie theft task - both feature based and LLM based.*

We believe our study offers several novel contributions that advance the field, which we listed in Introduction (lines 121-142):

- Validated model generalisability on speech data collected in-residence: After training and evaluating model performance on a benchmark DementiaBank dataset, we demonstrated the generalisability and applicability of our ML approach in a real-world context using an independently collected speech dataset.
- Identified the linguistic features most predictive of cognitive impairment: We provide insights into the contribution of individual linguistic features, enhancing our understanding of the changes in language most predictive of cognitive impairment.
- Risk stratification: To further enhance the clinical applicability of model predictions, we implemented risk stratification thresholds to distinguish low, medium, and high risk of

cognitive impairment. This approach enables clinicians to prioritise individuals who require further diagnostic workup, facilitating timely interventions.

- Pathway for scalable in-home monitoring: Our proposed approach shows promise for integration with conversational technology at home for accessible and scalable monitoring of cognitive health from verbal interactions. This realisation could enable scalable, non-invasive screening in community care settings.

We would like to thank the reviewers again for their time and help to improve our work.

Yours sincerely,

Maria R. Lima, on behalf of all co-authors.

References

- [1] Becker, J.T., Boiler, F., Lopez, O.L., Saxton, J. and McGonigle, K.L., 1994. The natural history of Alzheimer's disease: description of study cohort and accuracy of diagnosis. *Archives of neurology*, 51(6), pp.585-594.
- [2] Qian, W., Schweizer, T., Munoz, D. and Fischer, C.E., 2016. O3-04-06: Misdiagnosis of Alzheimer's Disease: Inconsistencies Between Clinical Diagnosis and Neuropathological Confirmation. *Alzheimer's & Dementia*, 12, pp.P293-P293.
- [3] Pennebaker, J.W., Chung, C.K., Frazee, J., Lavergne, G.M. and Beaver, D.I., 2014. When small words foretell academic success: The case of college admissions essays. *PloS one*, 9(12), p.e115844.
- [4] Boyd, R.L. and Pennebaker, J.W., 2015. Did Shakespeare write Double Falsehood? Identifying individuals by creating psychological signatures with text analysis. *Psychological science*, 26(5), pp.570-582.
- [5] Jordan, K.N., Sterling, J., Pennebaker, J.W. and Boyd, R.L., 2019. Examining long-term trends in politics and culture through language of political leaders and cultural institutions. *Proceedings of the National Academy of Sciences*, 116(9), pp.3476-3481.
- [6] Calzà, L., Gagliardi, G., Favretti, R.R. and Tamburini, F., 2021. Linguistic features and automatic classifiers for identifying mild cognitive impairment and dementia. *Computer Speech & Language*, 65, p.101113.
- [7] Boyd, R.L., Ashokkumar, A., Seraj, S. and Pennebaker, J.W., 2022. The development and psychometric properties of LIWC-22. *Austin, TX: University of Texas at Austin*, 10, pp.1-47.
- [8] Botelho, C., Gimeno-Gómez, D., Teixeira, F., Mendonça, J., Pereira, P., Nunes, D.A., Rolland, T., Pompili, A., Solera-Ureña, R., Ponte, M. and de Matos, D.M., 2024. Tackling Cognitive Impairment Detection from Speech: A submission to the PROCESS Challenge. *arXiv preprint arXiv:2501.00145*.
- [9] Gómez-Zaragozá, L., Wills, S., Tejedor-Garcia, C., Marín-Morales, J., Alcañiz, M. and Strik, H., 2023. Alzheimer disease classification through asr-based transcriptions: Exploring the impact of punctuation and pauses. *arXiv preprint arXiv:2306.03443*.

- [10] Sanguedolce, G., Brook, S., Gruia, D.C., Naylor, P.A. and Geranmayeh, F., 2024. When whisper listens to aphasia: Advancing robust post-stroke speech recognition. In *Proc. of Interspeech* (pp. 1995-1999).
- [11] Zusag, M., Wagner, L. and Thallinger, B., 2024. CrisperWhisper: Accurate Timestamps on Verbatim Speech Transcriptions. In *Proc. Interspeech 2024* (pp. 1265-1269).
- [12] Capstick, A., Palermo, F., Zakka, K., Fletcher-Lloyd, N., Walsh, C., Cui, T., Kouchaki, S., Jackson, R., Tran, M., Crone, M. and Jensen, K., 2024. Digital remote monitoring for screening and early detection of urinary tract infections. *NPJ digital medicine*, 7(1), p.11.
- [13] Perez-Toro, P.A., Ferrante, F.J., Pérez, G.N., Slachevsky, A., Nöth, E., Schuster, M., Maier, A., Arroyave, J.R.O. and Garcia, A.M., 2024. A cross-linguistic test of automated speech and language analysis for detecting Alzheimer's disease: Machine learning evidence from English and Spanish speakers. *Alzheimer's & Dementia*, 20, p.e084717.
- [14] Melistas, T., Kapelonis, L., Antoniou, N., Mitseas, P., Sgouropoulos, D., Giannakopoulos, T., Katsamanis, A., Narayanan, S. and Demokritos, N.C.S.R., 2023. Cross-lingual features for alzheimer's dementia detection from speech. In *Proc. Interspeech 2023* (pp. 3008-3012).
- [15] Shakeri, A., Farmanbar, M. and Balog, K., 2025. MultiConAD: A Unified Multilingual Conversational Dataset for Early Alzheimer's Detection. *arXiv preprint arXiv:2502.19208*.
- [16] Collins, G.S., Moons, K.G., Dhiman, P., Riley, R.D., Beam, A.L., Van Calster, B., Ghassemi, M., Liu, X., Reitsma, J.B., Van Smeden, M. and Boulesteix, A.L., 2024. TRIPOD+ AI statement: updated guidance for reporting clinical prediction models that use regression or machine learning methods. *bmj*, 385.

Reviewer Response

We thank the reviewers for their thoughtful feedback. We appreciate that **R2** and **R3** found the revised manuscript to be clear and coherent. We are grateful for **R2**'s commendation of our efforts to improve transparency and reproducibility. We are encouraged by **R3**'s recognition of the value of our independent pilot dataset and acknowledgement that this analysis validates the generalisability of our approach across a new cohort, language, and setting, *without* re-training or fine-tuning. The valuable feedback guided our improvements, highlighted in the revised manuscript in blue. A detailed response to reviewer comments, quoted in *italics*, follows.

Reviewer 2

The parallels between the current work and previous published work raise concerns about the novelty of the present submission. The authors' claim of novelty in generalizability, interpretability, and use of in-home data is not substantially distinct from what others have demonstrated. (...)

We thank the reviewer for this important clarification. We agree that previous work in the field – including Chandler et al. (2023) and Lindsay et al. (2021) – present contributions that target early screening from speech, clinical interpretability, and multilingual analysis. Our approach draws on these novel findings in placing greater emphasis on a path to *clinical utility*. Specifically, our work offers:

Independent validation using our parameterised model without re-training: We evaluated our best-performing model unchanged (i.e., *without* re-training or fine-tuning) on an independent pilot dataset to assess its generalisability across a different cohort, language, and setting. Results suggest the model captured transferable knowledge and could operate as an out-of-the-box solution, reflecting out-of-distribution performance closer to deployment than cross-validation alone. In contrast, Chandler et al. relied solely on leave-one-out cross-validation within a single dataset of 91 people.

Actionable triage framework: In conjunction with our clinical team (including neurologists and psychiatrists), we introduce a traffic-light stratification system that categorises model predictions into *clinically actionable* risk scores. Thresholds were defined on the validation set using ROC-AUC and Youden's J index to optimise both sensitivity and specificity. This approach aims to enhance clinical utility by reducing false positives and streamlining the triage process. Chandler et al. (2023) present an elegant dashboard showing PCA clusters and top feature distributions from decision tree ensembles for interpretability. Such a descriptive post-hoc tool explains *why* a class was predicted, but it does not operationalise model outputs into a clinical decision framework. Our contribution lies in converting predicted probabilities into actionable risk groups: high-risk cases (Red) can be acted on immediately with efficient allocation of medical resources, while uncertain cases (Amber) can be flagged for further diagnostic workup. This advance is specifically directed by clinicians as a basis to introduce the screening capacity into the treatment pathway.

Fairness and performance across different demographic and cognitive groups: We report subgroup performance analyses by sex, age and MMSE splits, demonstrating parity and highlighting expected uncertainty in MCI cases. Of particular importance, we consider the fact

that women are twice as likely to develop Alzheimer's disease as men [1] and report that our chosen model maintains robust predictions across male/female group splits (see details in Supplementary Material 10). To the best of our knowledge, such fairness analyses, which we consider essential when applying ML approaches in clinical pathways, have not been reported in prior work.

Predicted state of cognitive decline for prognostic assessment: Beyond classification, we model the state of cognitive decline by predicting MMSE as a continuous outcome and further validate generalisability of the chosen regressor model on the pilot dataset, again without re-training or fine-tuning (i.e., the model trained on DementiaBank data was exclusively used to infer state of cognitive decline on this pilot dataset), which offers a base for use in clinic. Given the progressive nature of ADRD, we see particular clinical utility in using speech to analyse trajectories of decline in a continuous spectrum. This approach provides a foundation for monitoring and prognostic assessment from longitudinal speech, which we suggest as future work.

Overall, our study offers a comprehensive, unified and clinically-oriented pipeline that integrates: i) automated screening of cognitive impairment from spoken language with external validation of our parameterised model unchanged on an independent dataset collected out-of-clinic including non-English speakers, ii) a risk stratification approach for clinically actionable triage, iii) fairness analysis across demographic subgroups alongside calibration of predicted risk with severity of cognitive impairment, and iv) continuous prediction of cognitive state to support future longitudinal studies.

We apologise for any lack of clarity on this point and have revised the manuscript to reflect these contributions in comparison to previous literature (lines 128-152, lines 555-566, lines 656-669).

Reviewer 3

(...) the manuscript could benefit from citing prior studies that have applied interpretable machine learning models to the detection of cognitive decline, particularly those addressing the more difficult three-way classification (ADRD, MCI, CN). (...) Similarly, the authors might consider citing accuracy results from the top teams in ADReSSo Challenge for comparative purposes.

We thank the reviewer for this valuable suggestion. We agree that situating our work in relation to interpretable machine learning studies and benchmark challenges strengthens the context. We have added relevant citations to ML studies on multi-class classification (lines 114-116) and clarified comparative performance results from studies using the ADReSSo dataset from DementiaBank (lines 515-519, lines 544-546).

We sincerely thank the reviewers again for their time and feedback. This input has guided revisions that have resulted in clearer presentation and delineation of the contributions of the investigation.

Yours sincerely,

Maria R. Lima, on behalf of all co-authors.

References

[1] Moutinho S. Women twice as likely to develop Alzheimer's disease as men-but scientists do not know why. *Nature medicine*. 2025 Mar;31(3):704-7.